# Multiple Functions of B Cells in the Pathogenesis of Systemic Lupus Erythematosus

**DOI:** 10.3390/ijms20236021

**Published:** 2019-11-29

**Authors:** Kongyang Ma, Wenhan Du, Xiaohui Wang, Shiwen Yuan, Xiaoyan Cai, Dongzhou Liu, Jingyi Li, Liwei Lu

**Affiliations:** 1Department of Rheumatology and Immunology, Second Clinical Medical College of Jinan University, Shenzhen People’s Hospital, Shenzhen 518000, China; kongyang@hku.hk (K.M.); liu_dz2001@sina.com (D.L.); 2Department of Pathology and Shenzhen Institute of Research and Innovation, The University of Hong Kong, Hong Kong 999077, China; duwenhan@hotmail.com (W.D.); xhwangbio@hotmail.com (X.W.); 3Department of Rheumatology, Guangzhou First People’s Hospital, School of Medicine, South China University of Technology, Guangzhou, 510000, China; 13631481240@163.com (S.Y.); caixytracy@126.com (X.C.); 4Department of Rheumatology and Immunology, Southwest Hospital, The First Hospital Affiliated to The Army Medical University, Chongqing 400038, China

**Keywords:** systemic lupus erythematosus (SLE), autoreactive B cells, age-associated B cells (ABCs), regulatory B cells (Bregs)

## Abstract

Systemic lupus erythematosus (SLE) is an autoimmune disease characterized by excessive autoantibody production and multi-organ involvement. Although the etiology of SLE still remains unclear, recent studies have characterized several pathogenic B cell subsets and regulatory B cell subsets involved in the pathogenesis of SLE. Among pathogenic B cell subsets, age-associated B cells (ABCs) are a newly identified subset of autoreactive B cells with T-bet-dependent transcriptional programs and unique functional features in SLE. Accumulation of T-bet^+^ CD11c^+^ ABCs has been observed in SLE patients and lupus mouse models. In addition, innate-like B cells with the autoreactive B cell receptor (BCR) expression and long-lived plasma cells with persistent autoantibody production contribute to the development of SLE. Moreover, several regulatory B cell subsets with immune suppressive functions have been identified, while the impaired inhibitory effects of regulatory B cells have been indicated in SLE. Thus, further elucidation on the functional features of B cell subsets will provide new insights in understanding lupus pathogenesis and lead to novel therapeutic interventions in the treatment of SLE.

## 1. Introduction

The immune system recognizes pathogens from the external environment and initiates protective immune responses, but ignores antigens derived from the host to maintain immune tolerance [1]. However, the breakdown of immune tolerance triggers the immune system to attack self-antigens and leads to autoimmune disorders [2,3]. Systemic lupus erythematosus (SLE) is a severe autoimmune disease, in which anti-nuclear antibodies (ANAs) and autoreactive B cells play crucial roles during disease progression and target organ damages [3,4,5,6].

Although numerous genetic predispositions and environmental cues have been identified in the pathogenesis of SLE [7,8], the generation and underlying molecular mechanisms of autoreactive B cells remain largely unclear. Age-associated B cells (ABCs), a novel B cell subset characterized by surface CD11c and transcription factor T-bet expression, are prone to secrete abundant autoantibodies during the pathogenesis of systemic autoimmunity, infection, and aging process [9,10,11]. In addition, innate-like B cells, including marginal zone (MZ) B cells and B-1 cells, express specific B cell receptors (BCRs) with high polyreactivities that can recognize self-antigens [12,13,14]. In mice, most B-1 cells reside in body cavities [15]. Notably, peritoneal cavity provides survival niches for B-1 cells during anti-mouse CD20 immunotherapy [16]. Moreover, the terminally differentiated long-lived plasma cells in SLE secrete a large amount of autoantibodies and become resistant to conventional immunosuppressive drugs [17].

In addition to autoantibody secretion, B cells perform multiple functions in the immune system, including antigen presentation and cytokine secretion [18]. In recent clinical studies on SLE patients, B cell depletion therapy with rituximab (anti-CD20) failed with a lack of significant clinical benefits during several randomized clinical trials [19,20]. Moreover, SLE patients with multiple courses of rituximab treatment even developed rituximab resistance, indicating the complexity of abnormal B cell-mediated immunity [21,22]. Increasing evidence indicates the involvement of regulatory B cells (Bregs) in maintaining immune homeostasis [23]. Bregs are defined as B cells with immunosuppressive function via producing inhibitory cytokines such as IL-10 and IL-35 [24,25,26]. The impaired function of Bregs has been observed during lupus development. Current studies have suggested that both impairments of Breg cell functions and expansions of autoreactive B subsets lead to immune tolerance breakdown and autoimmune progression [27]. In this review, we summarize recent progress in the identification and characterization of both pathogenic and regulatory B cell subsets in the development of SLE.

## 2. Age-Associated B Cells

In 2010, Mellor and colleagues firstly reported a distinct B cell subset with the co-expression of a dendritic cell marker CD11c, the immunoglobulins (IgM, Igκ, IgD), an intracellular enzyme indoleamine 2,3-dioxygenase (IDO), and the B cell lineage transcription factor the paired box 5 (PAX5). Upon toll-like receptor 9 (TLR9) ligation, this B cell subset showed T cell stimulatory properties as antigen-presenting cells (APCs) [28]. Subsequently, this unique B cell subset was named as age-associated B cells (ABCs) and found to be expanded during SLE pathogenesis in both mouse and human [29]. ABCs display distinct characteristics and transcriptional profiles when compared with follicular (FO) and MZ B cells. Currently, the generation and activation of ABCs together with their underlying molecular mechanisms have attracted extensive attention in studying the pathogenesis of SLE.

In 2011, Cancro and colleagues reported a distinct mature B cell subset without surface CD21/35 and CD23 expression. The CD21/35^−^ CD23^−^ B cells were resistant to BCR and CD40 ligation, but sensitive to TLR7 or TLR9 stimulation. As revealed by adoptive transfer experiments for studying the origin of ABCs, transfer of CD23^+^ follicular B cells reconstituted the CD21/35^−^CD23^−^ B subset in the recipient mice [30]. Notably, Marrack’s group also reported the increased population of CD11c^+^CD11b^+^ B cells in spleens of elderly female mice and lupus strains (female NZB/W F1, female Mer^−/−^, and male BXSB mice). Moreover, CD11c^+^CD11b^+^ B cells from aged mice and lupus-prone NZB/W mice both produced large amounts of anti-chromatin antibodies upon TLR7 ligation. Reconstitution of B cell subsets with the bone marrows from μMT and CD11c-DTR-GFP mice further indicated the importance of CD11c^+^CD11b^+^ B cells in anti-Smith IgG secretion [29].

Early studies revealed abnormal peripheral blood B cell homeostasis and CD21^-^CD38^-^ B cell expansion in patients with SLE [31]. Interestingly, Rothstein’s group detected a small CD11b^+^ B-1 cell subpopulation with CD11c and CD14 expression in SLE patients [32]. Human ABCs highly expressed surface markers such as CD20, CD32, CD95, CD75, FcRL, and IL-21 receptor, but showed low levels of CD21 and CD24 expression. Moreover, the increased proportions of the CD86^+^ subset within CD11b^+^CD11c^+^ B cells were also observed with increased T cell-stimulating activity in peripheral blood mononuclear cells (PBMCs) from SLE patients [33]. Recent studies found a predominant expansion of CD11c^+^T-bet^+^ B cell lacking surface IgD, CD27 and CXCR5 expression in PBMCs of African-American SLE patients with elevated serum anti-Smith and anti-RNA autoantibodies levels and nephritis development [33]. Together, these findings indicate an important pathogenic role of ABCs in lupus mice and SLE patients (Figure 1).

### 2.1. The Functional Features of ABCs

In chronic graft versus host disease (cGVHD)-induced lupus mice, elevated serum levels of anti-chromatin IgG2a and increased ABCs were observed [34]. The high levels of CD138 expression in ABCs indicate that ABCs were at pre-plasma cell differentiation stage with abundant anti-chromatin IgG2a secretion. Depletion of CD11c^+^ B cells or T-bet deficiency in B cells alleviated serum anti-chromatin IgG2a levels in cGVHD-induced lupus mice. Compared with naïve B cells, ABCs showed increased levels of Blimp1 expression in SLE patients [34].

TLR7 ligation has been shown to trigger the activation of pMAPK p38 and upregulation of CD86 and HLA-DR in ABCs [34]. Although B cells are known to produce a variety of cytokines, CD11c^+^ ABCs in SLE patients showed reduced capacity in secreting proinflammatory cytokines, such as IL-6 and TNF-α [35]. ABCs also displayed apoptotic signatures with downregulation of antiapoptotic gene *Bcl2* and upregulation of apoptotic gene *Fas* expression [35,36]. Consistently, CD11c^+^ B cells from SLE patients showed reduced cell viability and increased caspase-3 activity in culture [35]. 

Functional analysis suggests the antigen-presenting function of ABCs by in vitro and in vivo studies. However, Zhang and colleagues reported the reduced capacity to stimulate CD4^+^ T cell proliferation in T-bet^+^ CD11c^+^ B cells from SLE patients [37]. In response to CCL21 and CCL19, highly expressed chemokine CCR7 drove ABCs to reside at the T/B cell borders in spleens [38]. Moreover, the renal-infiltrated ABCs highly expressed the chemokine receptor CXCR4, which might be involved in the recruitment of ABCs to the inflamed kidney [39].

### 2.2. The Transcriptional Network in ABCs

The Th1 lineage transcription factor T-bet (encoded by *Tbx21* gene) is essential for the generation of ABCs, which was supported by the recent findings that conditional depletion of *Tbx21* in B cells significantly abrogated the generation of CD11c^+^ B cells in lupus mice, along with lower serum antibody levels and ameliorated renal damages [40].

Mechanistic studies revealed the activation requisites and regulatory cues during ABC generation. The ligations of TLRs, together with the stimulations of cytokines such as IL-21 and IFN-γ, induced the generation of ABCs. In B cell culture system, IFN-γ directly activated T-bet expression in the presentence of TLR engagement. Moreover, IL-21 triggered the expression of both T-bet and CD11c in B cells of mice and humans with TLR engagement [35,41,42]. Accumulated CD11c^+^T-bet^+^ ABCs were observed in IL-21 transgenic mice. In contrast, IL-4 antagonized the induction of T-bet in B cells [41,42].

Ligations of TLRs and stimulation of cytokine were also found to induce the differentiation of naïve B cells into ABCs from PBMCs of SLE patients or healthy donors [33]. Moreover, the increased somatic mutation in ABCs might result from high expression levels of activation-induced cytidine deaminase (AICDA), since the knockdown of *Tbx21* in B cells significantly reduced the mRNA levels of *Aicda* transcript [43]. Moreover, TLR7/9 and Myd88 signal pathways were also required for the expansion of ABCs [29]. TLR9 immune complex, composed of a biotinylated CpG-rich dsDNA fragment (TLR9 agonist) and a BCR ligand, triggered the proliferation and mitochondrial apoptosis in B cell subsets, respectively. In addition, TLR9 immune complex, in combination with anti-CD40, IL-21, or IFN-γ, enhanced ABC generation in cultured B cells [44]. Previous studies also indicate the importance of JAK/STAT signal in the induction of ABCs. STAT1- or STAT4-knockout splenocytes failed to express T-bet, which might be due to the impaired IFN−γ secretion [41,45,46].

Although follicular helper T cells (Tfh) and Tfh-derived cytokine IL-21 have been shown to promote the generation of ABCs, Th1 cells also play crucial roles in driving ABC differentiation [47,48,49]. Recent studies using single-cell RNA sequencing technology to examine renal biopsy samples from lupus nephritis revealed significant local infiltration and activation of ABCs in SLE patients with renal damages. Several ABC-related genes are upregulated in inflamed kidney, such as *Tbx21*, *Fas*, *Tbk1*, *Irf4,* and *Il21*. These renal-infiltrated ABCs showed unique gene expression pattern, including high expression levels of *Tbx21, Tlr7,* and *Zeb2,* but low expression levels of *Traf5* and *Zeb1* [39]. As a major regulator for EGR, ATF3 was highly expressed in ABCs and also showed the highest levels of DNA accessibility in ABCs within B subsets in SLE patients [50,51].

Currently, it is largely unclear how ABCs are functionally regulated. The SWEF family members, SWAP-70 and DEF6A, belonging to Rho GTPase–regulatory proteins, are shown to control the activity of interferon-regulatory factors (IRFs) and modulate the generation of ABCs [52]. Importantly, the DEF6 locus has been identified as a genetic risk factor in SLE [53]. Knockout of *Swap-70* and *Def6* (double-knockout, DKO) led to the spontaneous expansion of ABCs and development of lupus in C57BL/6 mice. As both SWAP-70 and DEF6 could modulate the activation of IRF4 and IRF5 [54,55,56], further analysis of specific deletion of *Irf4* in DKO CD11c-expressing cell (Cd11c-Cre Irf4^fl/fl^ DKO mice) showed no significant changes in ABC expansion or autoantibody secretion. However, specific deletion of *Irf5* in DKO B cells (Cd21-Cre Irf5^fl/−^ DKO mice) resulted in the absence of ABC cells and ameliorated lupus development compared with DKO mice [57].

Since the *Irf5* gene variants are strongly associated with serum IgG2a and IL-6 levels in patients with SLE [58], the specific binding of IRF5 within *Il6* transcription start site (TSS), the *Ighg2c* region and Jun protein was observed in DKO B cells during IL-21 stimulation. SWAP-70 and DEF6 could bind to the IRF-association domain of IRF5 and competitively modulate the binding of T-bet in target genes [59]. The lack of SWEF proteins led to enhanced IRF5 activities in response to IL-21 stimulation and the activation of T-bet in DKO mice. The generation of ABCs in DKO mice was dependent on the T/B cell interaction and IL-21 secretion. Depletion of IL-21 dramatically reduced the expansion of ABCs, follicular helper T cells (Tfh), germinal center (GC) B cells, and plasma cells (PC) in DKO mice [57]. Signaling lymphocyte activation molecule-associated protein (SAP, encoded by *SH2D1a*) controls the interaction between Tfh cells and B cells within GC structure [60]. Depletion of *SH2D1a* in DKO B cells significantly impaired the accumulation of ABCs, Tfh cells, GC B cells, and PCs [59]. Moreover, age-associated infusion of thymic B cells with T-bet expression, IgG2a secretion, and diminished autoimmune regulator (AIRE) expression controlled the central T cell tolerance during aging and autoimmune disorders [47]. Although the *Aire* gene mutations lead to autoimmune polyendocrinopathy candidiasis ectodermal dystrophy (APECED) [61], the mechanism of AIRE downregulation and T-bet upregulation in thymic ABCs remains uncharacterized.

## 3. Innate-Like B Cells

The innate-like B cells include B-1 cells and MZ B cells [14,62,63,64]. These innate-like B cells express high levels of toll-like receptors (TLR2, TLR4, and TLR9) and are responsive to both pathogens and endogenous ligands via antibody and cytokine secretion [42,65,66].

A restricted set of B-1 cells expressed the T15 idiotype that could bind to the phosphorylcholine (PC) and exhibited a protective function against infections. However, the T15 idiotype also recognized oxidized low-density lipoprotein (oxLDL) and apoptotic cells [67,68]. MZ B cells are located in MZ regions and produce a polyreactive antibody encoded by M167 idiotype with similar specificity to T15 [69]. The CD23^lo^CD21^hi^ MZ and CD5^+^ B-1 B cells respond to T cell-independent antigens and undergo polyclonal expansion upon stimulation, as revealed by the findings that MZ B cells and B-1 cells rapidly produce a large amount of IgM during the initial stages of bacterial infection in mice (Figure 2) [12].

### 3.1. The Functional Features of Innate-Like B Cells

Several reports have shown significantly increased B-1 cells in peritoneal cavities, spleens, and kidneys of lupus-prone NZB/W and B6.*Sle2* mice. Depletion of B-1 cells significantly ameliorated lupus progression in mice [13,72,73]. Recent studies further characterized the CD20^+^CD43^+^CD27^+^CD70^−^ human B-1 cell subset in the umbilical cord and adult peripheral blood [74]. The expanded CD11b^+^CD11c^+^ B cells in PBMCs of SLE patients observed in early studies appeared to be B-1 cells, which was further supported by more recent findings that the frequencies of B-1 cells were markedly increased in SLE patients [32].

The expansion of autoreactive MZ B cells was detected in several lupus-prone mouse models such as NZB/W F1, B6.*Sle1Sle2Sle3,* and BAFF-transgenic mice [75,76]. Adoptive transfer of MZ B cells accelerated lupus progression [77]. In B6.*Sle1Sle2Sle3* mice, a large portion of autoreactive MZ B cells entered into lymphoid follicles before the onset of lupus nephritis [77]. Upon the persistent contaction of CD4^+^ T cells, intrafollicular autoreactive MZ B cells showed enhanced proliferative capacity [77]. Although MZ B cells contribute to the progression of lupus in mice, both phenotypic features and functional properties of MZ B cells need to be further investigated in humans. Apart from their autoreactive effects, innate-like B cells also produced IL-10 during autoimmune diseases [78].

### 3.2. The Transcriptional Network of Innate-Like B Cells

BCR signal pathways positively modulate the function of innate-like B cells [79]. Deficiency of BCR signal inhibitory coreceptors CD22 and Sialic acid-binding immunoglobulin-type lectin-G (Siglec-G) led to the significant expansions of B-1 cells and MZ B cells, together with lupus-like systemic autoimmune development in mice [80]. Siglec-G was highly expressed in B-1 cells [81], depletion of Siglec-G alone triggered the amplification of B-1 cells in the spleen and peritoneal cavity [80]. The Siglec-G-deficient B-1 cells showed enhanced capacities for natural IgM antibody secretion and stronger calcium signal compared with wild-type B-1 cells [81].

NF-κB pathway is downstream of BCR signal [82], and Ikaros supports the sustained binding of NF-κB within chromatin [83]. Specific deletion of *Ikaros* altered the homeostasis of B cells by amplifying the innate-like B cells (B-1 cells and MZ B cells) and suppressing the follicular B cell (Fo B) generation. Importantly, the depletion of *Ikaros* in B cells triggered autoantibody secretion and lupus nephritis development in mice. *Ikaros* depletion modulated serval gene expressions in B-1 cells, such as the upregulation of *Nrp2, Teme176b, Gna15,* and *Slamf9* and downregulation of *Slpi, Tyrobp, Prib,* and *Rapgef4* genes [84]. Moreover, the upregulation of orphan nuclear hormone receptor Nur77 was also observed in lupus B-1 cells [85]. These distinct functional features and transcriptome profiles may lead to pathogenic roles and autoreactivity of innate-like B cells in SLE.

## 4. Autoreactive Plasma Cells

During humoral immune response, activated mature B cells differentiate into memory B cells and plasma cells [86]. Both short-lived plasmablasts and long-lived plasma cells generate abundant autoantibodies in SLE [87]. However, long-lived plasma cells do not undergo DNA synthesis and cell proliferation. It becomes clear that the survival of plasma cells is critically dependent on the survival niches formed by stromal cells and cytokines in the bone marrow and inflamed organs [88]. Moreover, spleen and inflamed kidney could provide the survival niches to long-lived plasma cells in lupus-prone NZB/W mice [89]. The autoreactive long-lived plasma cells are resistant to conventional immunosuppressive drugs, such as dexamethasone and cyclophosphamide [90,91]. Thus, depletion of autoreactive long-lived plasma cells is crucial for developing effective therapies in SLE patients with target organ damages (Figure 3).

### 4.1. The Functional Features of Plasma Cells

The phenotypic features of autoreactive plasma cells are poorly characterized in SLE. Several studies used direct staining of fluorochrome-binding dsDNA to detect dsDNA-specific plasma cells in NZB/W mice [92,93,94]. In addition, a significant amplification of Lineage^-^CD27^+^CD38^+^CD138^+^ plasma cells was observed in PBMCs of active SLE patients. Although previous findings in plasma cells from the bone marrow showed the absence of TLR expression [95,96], we and others recently identified a novel subset of autoreactive long-lived plasma cells with surface TLR4 and CXCR4 expression in human and murine lupus [96,97].

In the inflamed kidney of lupus-prone NZB/W mice, significant infiltration of CD138^hi^MHCII^+^IgG^+^ plasma cells with dominantly IgG autoantibody-secreting ability was observed. Notably, adoptive transfer of plasma cells promoted lupus development in recipients. Moreover, the renal-infiltrated CD138^hi^MHCII^+^IgG^+^ plasma cells preferentially expressed CXCR3 [17]. Long-lived plasma cells have distinct metabolic properties that specifically modulated the cell survival [98]. Compared with short-lived plasma cells, long-lived plasma cells robustly engaged in glucose uptake and pyruvate-dependent respiration [98]. Therefore, these findings may facilitate the development of new approaches to target autoreactive PCs in the treatment of refractory SLE with target organ damages.

### 4.2. The Transcriptional Network of Plasma Cells

The development of plasma cells is modulated by distinct transcriptional programs [99]. Long-lived plasma cells are mainly generated from GC B cells and memory B cells [100]. The gene expression profiles of long-lived plasma cells are considerably different from GC B cells and memory B cells. Gene expression programs of downregulation in paired box protein Pax-5, microphthalmia-associated transcription factor (MITF), BTB domain and CNC homolog 2 (BACH2), B cell lymphoma 6 (BCL-6), Metastasis-associated 1 family, member 3 (MTA3) expression and upregulation of X-box-binding protein 1 (XBP-1), interferon-regulatory factor 4 (IRF4), B-lymphocyte-induced maturation protein 1 (Blimp1) occur during plasma cell generation [101]. Expression of PAX5 suppressed XBP-1 expression and MITF also inhibited the generation of plasma cells via negative regulation of Blimp1 expression [102,103,104]. Although proliferating plasmablasts, short-lived plasma cells, and long-lived plasma cells all showed high expression levels of CD138, IRF4, XBP-1 and Blimp1, the long-lived plasma cells did not express the cell dividing marker Ki-67 [105,106]. Shlomchik and colleagues used the BrdU and EdU pulse-labeling experiments to reveal the kinetics of antigen-specific long-lived plasma cell formation from germinal center response and memory B cells in vivo [107]. However, the distinct transcriptional profiles in autoreactive plasma cells with target organ homing capacities remain largely unknown. Recent genome-wide association studies have suggested that rs548234 is the SLE-associated risk allele of *PRDM1* (coding Blimp1) [108]. Depletion of plasma cells with proteasome inhibitor bortezomib effectively ameliorates lupus development in lupus mouse models and SLE patients [109,110].

## 5. Regulatory B Cells

Regulatory B cells (Bregs) are defined as B cell subsets with immunosuppressive function and are involved in various immuno-pathological processes. Bregs secrete large amounts of inhibitory cytokines, such as IL-10, TGF-β, and IL-35. In addition, Breg cells may exert their function by cell–cell contact-dependent manners via ligations of negative co-stimulators, such as FasL, GITRL, and PD-L1 [111].

It is reported that naive B cells, immature B cells, and plasma cells can differentiate into Bregs upon stimulation by microenvironmental factors during immune responses [23]. Various phenotypic features of Bregs have been reported in different disease models [112]. For instance, Tedder’s group has identified CD1d^hi^CD5^+^ B cells, namely, B10 cells, as the major IL-10-secreting B cell subset that possessed potent regulatory effects during the autoimmune response by inhibiting T cell responses [113,114,115]. Mauri’s group revealed that IL-10-secreting B cells were derived from CD1d^+^CD21^+^CD23^+^ transitional stage 2 and marginal zone precursor B cells (T2-MZP) in mice with collagen-induced arthritis (CIA) [116]. Gray et al. observed that apoptotic cells could induce the production of IL-10 in CD21^+^CD23^-^ MZ B cells [78]. Moreover, Kurosaki and colleagues reported that IL-10-producing B cells mainly exhibited the phenotype of CD19^+^CD138^+^ plasmablasts in the draining lymph nodes of mice with experimental autoimmune encephalomyelitis (EAE) and played crucial regulatory roles during EAE development [117]. Fillatreau and colleagues demonstrated that Bregs also secreted an inhibitory cytokine, IL-35 [118,119]. In addition, B cells expressing the inhibitory co-stimulatory molecule T cell Ig and mucin 1 (Tim-1) were reported to differentiate into IL-10-secreting Bregs upon the stimulation of Tim-1 ligands [120]. B cells with high levels of PD-L1 expression showed the regulatory capacity to inhibit antigen-specific humoral immune responses [121]. In mice with the experimental autoimmune hepatitis (AIH), Chu and colleagues observed the expanded CD11b^+^ regulatory B cells with the capacity of suppressing T helper cell response [122]. Furthermore, CD9 was identified as a unique functional marker for IL-10-producing regulatory B cells, since B cells deficient for CD9 showed impaired regulatory functions when compared with wild-type (WT) counterparts [123].

### 5.1. The Functional Features of Bregs

Both clinical and experimental studies have found diverse phenotypes of Bregs during the pathological processes of SLE [124]. Impaired immune suppressive function of human CD19^+^CD24^hi^CD38^hi^ Bregs was observed in SLE patients [125]. The CD19^+^CD24^hi^CD38^hi^ Bregs from healthy individuals significantly inhibited the production of IFN-γ and TNF-α by activated T helper cells via the release of IL-10 and the engagement of CD80/CD86. Although comparable numbers of CD19^+^CD24^hi^CD38^hi^ Bregs were observed between healthy individuals and SLE patients, CD19^+^CD24^hi^CD38^hi^ Bregs in lupus patients failed to suppress T helper cell differentiation and proinflammatory cytokine secretion, which might be due to the dampened IL-10 secretion and impaired CD40 signal [125].

In lupus-prone Roquin^san/san^ mice, the deficiency of IL-17A significantly abrogated nephritis development, together with increased numbers of CD1d^+^CD5^+^ Bregs, Foxp3^+^CD25^+^CXCR5^+^ follicular regulatory T cells (Tfr), but decreased numbers of CD4^+^ICOS^+^CXCR5^+^PD1^+^ follicular helper T cells (Tfh) and GC B cells [126]. Our previous studies also revealed the potent suppressive function of Bregs and Breg-derived IL-10 in Th17 differentiation during CIA development [127]. Moreover, adoptive transfer of CD1d^+^CD5^+^ Bregs significantly suppressed the generation of IL-17A-secreting Th17 cells and ameliorated CIA pathogenesis. Consistently, IL-10 deficiency in B cells abrogated the suppressive function of Bregs [127].

Recently, we reported a novel function of Bregs in modulating Tfh generation in patients with primary Sjögren’s syndrome (pSS) [128]. The decreased Bregs and increased Tfh cells were inversely correlated in both pSS patients and mice with experimental Sjögren’s syndrome (ESS). In co-culture experiments of T cells with either wild-type (WT) or *Il10^−/−^* B cells, B cell-derived IL-10 was shown to be critical in suppressing the differentiation and generation of Tfh cells [128]. Interestingly, a positive correlation between the frequencies of circulating CD1d^hi^CD5^+^ Bregs and CXCR5^+^PD-1^+^ Tfh was observed in PBMCs of SLE patients [129]. Further co-culture experiments revealed that both Tfh and Tfh-derived IL-21 significantly promoted IL-10 secretion and Breg differentiation, indicating the positive feedback of Tfh in modulating Breg generation [129]. Besides, IL-10-secreting Bregs also triggered the amplification of Foxp3^+^ Treg cells. Adoptive transfer of IL-10-secreting Bregs significantly inhibited lupus progression in lupus-prone MRL/Lpr mice [130].

In 2014, Fillatreau and colleagues identified IL-35-producing B cells as a key player in the negative regulation of autoimmunity [131]. B cell-specific depletion of IL-35 led to the activation of T and B cells during EAE development. It was found that IgM^+^CD138^hi^TACI^+^CXCR4^+^CD1d^int^ Tim1^int^ plasma cells with Blimp1 expression were the major source of IL-35 and IL-10 in spleens and draining lymph nodes of EAE mice [131].

### 5.2. The Regulation of Bregs

It has been shown that activation of BCR, Toll-like receptors (TLRs), CD40, Tim-1, and CD19 induces the differentiation of naïve B cells into Bregs [132]. Several proinflammatory cytokines such as B cell-activating factor (BAFF), IL-21, IL-6, IL-1β, IL-35, TGF-β, and type 1 interferons are found to promote the generation of Bregs. In addition, apoptotic cells and lipid antigens are involved in the differentiation of Bregs [133]. The shift of B cell activation, B cell co-stimulation, and cytokine signal pathways may dampen the regulatory functions of Bregs during SLE pathogenesis.

#### 5.2.1. B Cell Receptor (BCR)

The ligation of BCR by autoantigens is involved in the activation and IL-10 secretion in B cells [134]. Treatment of B cells with anti-IgM induces IL-10 secretion [120]. Although ligations of BCR increases intracellular calcium concentrations and activation of multiple signal pathways, such as NF-κB, ERK, Jnk, p38, and NFAT families, the precise mechanisms in modulating IL-10 secretion in B cells upon BCR ligation are not fully elucidated. B cell-specific deficiency for stromal interaction molecule 1/2 (Stim1^f/f^ Stim2^f/f^ Mb1^cre/+^) led to diminished IL-10 secretion upon BCR ligation [135]. Moreover, the immunosuppressive drug cyclosporine A (CsA), which inhibits the phosphatase calcineurin, also significantly suppressed IL-10 secretion in B cells upon BCR ligation [135].

#### 5.2.2. Toll-Like Receptors (TLRs)

TLRs are important pathogen sensors in the recognition of pathogen-associated molecular patterns (PAMPs), such as LPS and heat-shock proteins (HSPs) [136]. B cells are known to express elevated levels of TLRs, such as TLR2, TLR4, TLR7, TLR8, and TLR9, during the development of SLE [137]. It was reported that LPS (TLR4 ligand) and CpG-DNA (TLR9 ligand) both dramatically induced the generation of IL-10-secreting Bregs [138,139]. Adoptive transfer of LPS-activated B cells significantly ameliorated the disease procession in non-obese diabetic (NOD) mice [138]. In addition, TLR2 ligands, such as Pam3CSK4 and FSL1, and TLR7 ligand, such as imiquimod, also induced the secretion of IL-10 in B cells and triggered the generation of Bregs [140].

Myeloid differentiation primary response 88 (MyD88) is the major downstream molecular of TLRs [141]. Depletion of *Myd88* in B cells limited IL-10 secreting capacities of B cells both in vitro and in vivo. Moreover, one atypical IκB protein (IκBNS) also played important roles in TLR-ligation-induced Breg generation [140]. Mice deficient for *IκBNS* showed reduced frequencies of CD1d^+^CD5^+^ Bregs, along with the limited responsiveness of IL-10-secreting Breg generation upon TLR2-, TLR4-, TLR7-, or TLR9-ligation [140].

#### 5.2.3. Co-stimulatory Molecules

Both CD40 and Tim-1 are shown to modulate the generation of Bregs [142]. Mature B cells express CD40 that mediates B cell activation and differentiation [143]. In lupus-prone MRL/Lpr mice, successive administration of anti-CD40 antibodies remarkably induced the expansion of IL-10-secreting transitional B cells, along with ameliorated lupus nephritis development [142]. Ligation of CD40 also triggered the activation of STAT3 and led to IL-10 secretion in human B cells [142]. Moreover, Tim-1 is a membrane surface glycoprotein with immunomodulatory effects [144]. Recently, the inhibitory functions of Tim-1^+^ B cells were found to be mediated via the ligation of Tim-1 [120].

#### 5.2.4. Cytokines

Several cytokines are involved in the generation and differentiation of Bregs. Although the belimumab, a biologic drug for the blockade of BAFF, has been shown to inhibit the survival of autoreactive B cells in SLE [145], we found that low-dose BAFF treatment induced the generation of IL-10-secreting Bregs both in vitro and in vivo. Moreover, we showed that adoptive transfer of BAFF-induced Bregs significantly ameliorated CIA and ESS development [127,128].

In addition, CpG, anti-CD40, and type 1 IFNs are shown to induce IL-10-secreting CD24^hi^CD38^+^ Breg differentiation in PBMCs [125]. A marked expansion of IL-10-secreting CD24^hi^CD38^+^ Bregs was observed when co-cultured with plasmacytoid dendritic cells (pDCs) of healthy individuals, while blockade of IFN-α and CD40 abrogated pDC-triggered Breg expansion [146]. In active SLE patients, the frequencies of pDCs showed a dramatic decrease. Moreover, pDCs from healthy individuals significantly promoted the expansion of Bregs when compared with pDCs from SLE patients [146], indicating that the impaired pDCs function in lupus patients may attribute to the reduced generation of Bregs. Notably, the treatment with antibiotics also decreased the generation of IL-10-secreting T2-MZP Bregs in mice [147]. The gut microbiota promoted IL-10-secreting Breg generation in the spleen and mesenteric lymph nodes via the release of IL-1β and IL-6. Recent studies have demonstrated that B cell-specific deficiencies of IL-1 receptor 1 (IL-1R1) or IL-6 receptor (IL-6R) impair the generation of IL-10-secreting B cells and exacerbate inflammatory responses during arthritis development [147]. Furthermore, cytokines IL-35 and IL-33 can trigger the expansion of Bregs [148]. Treatment with IL-35 significantly increased the numbers of IL-10-secreting CD1d^+^CD5^+^ Bregs and ameliorated renal damages in lupus-prone MRL/Lpr mice [146].

## 6. Perspectives

During last several decades, various types of therapeutics have been developed for the treatment of SLE. Conventional immune suppressive drugs such as cyclophosphamide eliminate activated B cells undergoing cell proliferation, whereas rituximab and epratuzumab, monoclonal antibodies against CD20 and CD22, are used to deplete B cells in human and murine lupus [149,150]. However, epratuzumab failed to reduce serum autoantibody levels in SLE patients [151,152]. Inhibitions of T/B cell co-stimulatory pathway such as CD40-CD40L lead to the effective improvement of disease progression in lupus mice, but unexpected thromboembolic events have been reported during the pre-clinic trials of anti-CD40L antibodies. The CTLA4 fusion protein (CTLA4Ig) has been reported to ameliorate lupus nephritis development in mice, while their phase I/II clinic trails have been underway in SLE patients. The proteasome inhibitors and anti-CD138/CD38 antibodies are shown to target autoreactive plasma cells in SLE, and we recently found that proteasome inhibitors can target proinflammatory Th17 cells in mice with ESS [110,153,154,155,156]. There is compelling evidence that blockades of cytokines, including BAFF, BCMA, APRIL, IL-6, and/or IL-21, inhibit B cell proliferation in murine lupus [157]. Belimumab, a neutralizing antibody against human BAFF, has been recently approved for the treatment of active SLE patients. Blockades of TLR7/9 ameliorate lupus nephritis development, which may be partially due to the depletion of autoreactive B cells [158,159]. Moreover, blockades of CXCR4, CXCR3, and other chemokine interactions may abrogate the migration of autoreactive B cells and plasma cells towards inflamed organs [160,161]. Thus, further studies on the homeostatic regulation and functional interactions between various B and T cell populations will provide new insights into the effective treatment of SLE [128,162].

Currently, the identified targets in autoreactive B cells such as CD11c and T-bet may facilitate the development of B cell-targeted therapies in refractory SLE with organ damages. Correspondingly, further elucidations of the immunosuppressive functions of Bregs will provide novel strategies for Breg-targeted immunotherapy in SLE patients. Although various phenotypes of Bregs have been reported, such as CD24^hi^CD38^hi^, CD24^hi^CD27^+^, CD25^hi^CD71^hi^CD73^lo^, and CD19^+^Tim1^+^ B subsets, the suppressive functions of Bregs via secretion of IL-10 and IL-35 have highlighted a regulatory role of Bregs in maintaining immune homeostasis in human. Future studies on the impaired inhibitory effects of Bregs may lead to the clinical applications of Breg cell therapy in SLE [163]. Thus, a combination of in vitro expansion of Bregs and adoptive transfer of autologous Bregs may restore the immune regulatory functions of Bregs and effectively suppress effector T cell response in SLE [164].

Although dysregulations of pathogenic and regulatory B subsets have been characterized during lupus development, B cell-targeted therapy in SLE also raises the concern on potential side effects and clinic relapses due to the possible deletion of regulatory B cells. However, the cell therapy of autologous Breg transfer may offer a promising strategy for the treatment of SLE, as reflected by our recent findings that transfer of in vitro expanded IL-10-producing Bregs can effectively ameliorate the disease progression in mice with CIA or ESS [127,128]. Increasing evidence indicates the plasticity of T subsets and disturbance in Treg/Th17 balance during autoimmune pathogenesis [165,166], but how B cell subsets undergo functional differentiation in inflammatory milieu during SLE development remains largely unclear. Further investigations on the potential plasticity of B cells in vivo may help improve the efficacy of B cell-targeted therapy in clinical applications. Future studies will need to elucidate the mechanisms underlying the imbalance between autoreactive B cells and Breg subsets during the pathogenesis of SLE by identifying the new functional features of B cell subsets and searching for novel therapeutic targets in the treatment of SLE.

In summary, a fuller understanding of the features of autoreactive B cells and Breg subsets in SLE will enhance the clinical applications of new biomarkers and validate novel therapeutic targets in SLE.

## Figures and Tables

**Figure 1 ijms-20-06021-f001:**
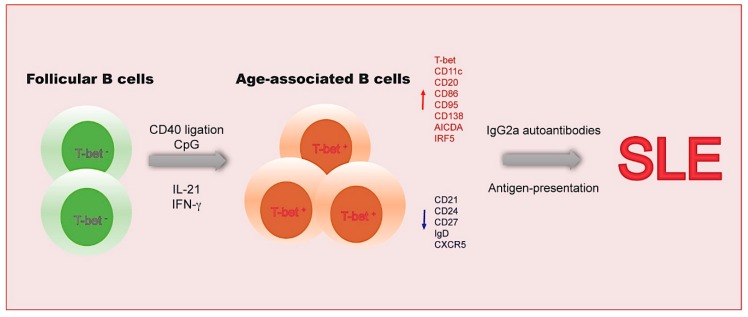
Age-associated B cells in systemic lupus erythematosus (SLE). Age-associated B cells (ABCs) are a novel subset of autoreactive B cells characterized by their unique expression of CD11c and the transcription factor T-bet. During the stimulation of anti-CD40, CpG, IL-21, and IFN-γ, naïve mature follicular B cells differentiate into CD11c^+^T-bet^+^ ABCs. Importantly, significant accumulation of ABCs accelerates SLE progression via autoantibody secretion and autoantigen-presentation.

**Figure 2 ijms-20-06021-f002:**
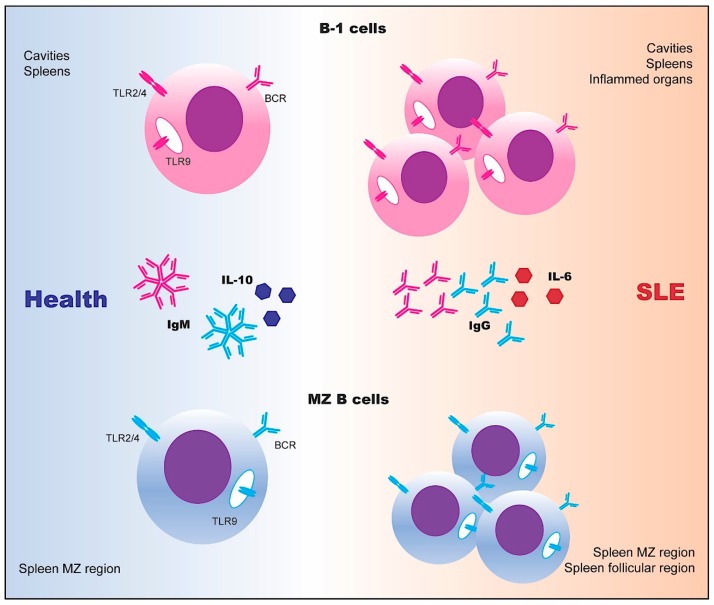
Innate-like B cells in SLE. Marginal zone (MZ) B cells and B-1 cells have the potential to generate cross-reactive and autoreactive B cell receptors (BCRs) that recognize epitopes on apoptotic cells. In healthy conditions, B-1 cells and MZ B cells secrete IgM and regulatory cytokine IL-10 to maintain immune tolerance. However, the activated and dramatically expanded autoreactive B-1 cells and MZ B cells in SLE produce large amounts of IgG autoantibodies and proinflammatory cytokines, such as IL-6 [12,70,71].

**Figure 3 ijms-20-06021-f003:**
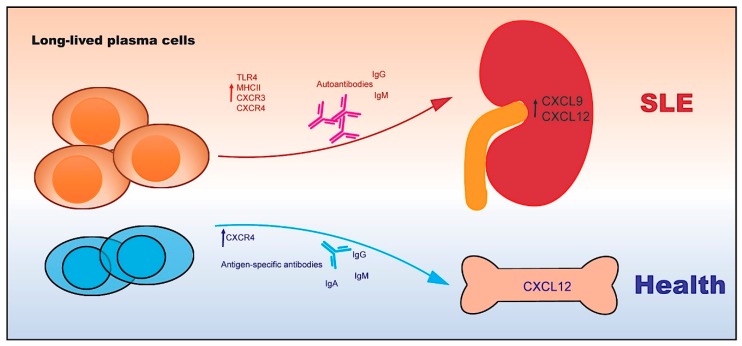
Long-lived plasma cells in SLE. The terminally differentiated long-lived plasma cells secrete large amounts of antibodies. In the healthy condition, the majority of long-lived plasma cells may migrate to bone marrow survival niches. However, the upregulated CXCL9 and CXCL12 in the inflamed organs of SLE, such as kidney, may recruit the autoreactive long-lived plasma cells to maintain local autoimmune inflammation.

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
