# Peer review of "Multiple Functions of B Cells in the Pathogenesis of Systemic Lupus Erythematosus"

_ijms, 2019, doi:10.3390/ijms20236021_

Round 1
Reviewer 1 Report
This is a very well done review by a group of scientists working on lymphocyte development and its dysregulation in autoimmune diseases.
Potential, the role of B cells in the systemic lupus erythematosus will lead to the identify new targets to help with the prophylaxis and therapy of this disease.
Extra paper related to the topic and a few clinical trials were found (Pls see below). This reviewer believes that the authors should elaborate better the last point 6 adding these papers and clinical trials.
The authors at the end mentioned the potential applications, but it should be nicer to compare with current few clinical trail and more specific how these multiple functions of B cells described could impact the future clinical trials.
Drugs. 2006;66(15):1933-48.
B-cell-targeted therapy for systemic lupus erythematosus.
Sabahi R1, Anolik JH.
BioDrugs. 2008;22(4):239-49.
B-cell-targeted therapy for systemic lupus erythematosus: an update.
Ding C1, Foote S, Jones G.
Author information
BioMed Research International
Volume 2019, Article ID 7948687, 7 pages
https://doi.org/10.1155/2019/7948687
Review Article
Research Progress on Regulatory B Cells in Systemic Lupus Erythematosus
Tao Wang, Yongjun Mei, and Zhijun Li
Editor's Choice
B-cell therapy in lupus nephritis: an overview
Salem Almaani, Brad H Rovin
Nephrology Dialysis Transplantation, Volume 34, Issue 1, January 2019, Pages 22–29, https://doi.org/10.1093/ndt/gfy267
FEW CLINICAL TRIALS are asking whether could Targeted Elimination of B-cell Depletion Therapy and/or Combination Therapy Restore Peripheral B Cell Abnormalities in Systemic Lupus Erythematosus(SLE)
Author Response
We appreciated the reviewer’s insightful comments. In line with the reviewer’s suggestions, we have included the recommended references and discussed recent advances in B-cell targeted therapy and the impact of multiple functions of B cells in the revised text (please refer to lines 531-547 and 567-582, reference 151-153 and 164). All line numbers are shown in track changes with ‘All Markup’ view.
Reviewer 2 Report
Dear Editor,
The paper addresses the issue of some relevant pathologies of B-cells in SLE. It focuses on certain subsets, with sufficient information about the general knowledge about these subtypes, and intends to link them to SLE pathology, with the aim to foster future research. The information given is correct, well-structured and relevant, with some fluctuations within the text both in terms of relevance SLE and English grammar.
Line 188: "The CD23loCD21hiMZ and CD5+ B-1 B cells respond to T cell-independent antigens and undergo polyclonal expansion during multiple stimuli". This is a very general sentence, and no further detailed description of this process is presented, rather a schematic Figure is presented, but it is not clear, from what research data (citations, own previous experiments) does the information presented in Figure 2 comes from. A more clear description of the proposed mechanism and sources of the data is needed, also specifying if these are animal or human data. Line 226. Is the PIK3CD-E1020K model an SLE model or of some other autoimmune disease? Line 230-231: The first half of the sentence describes the role of IPK3CDE in B-cell differentiation, and the second half concludes that this indicates an important pathogenic roles of innate-like B cells in SLE. The causal relationship is not clear, and further data supporting this important role should be presented. What functional sequelae of this skewed B-cell differentiation have been published, in general, or related to SLE? Line 274. The section 4.2 contains no SLE-related information, only general knowledge about plasma cell physiology. Since autoantibodies are crucial to the pathogenesis of SLE, it has long been obvious that plasma cells are important players of SLE pathogenesis. The Chapter 4. therefore should contain novel and lupus-related information relevant to the whole concept of the paper, or this should be omitted. Line 314."CD9 was identified as a unique functional marker for IL-10-producing regulatory B cells". Does it mean that CD9 is a B-reg specific-marker, or it is expressed in a wider variety of cells, but its function is most important in B-regs? Line 358-: The sectoin 5.2 The regulation of Bregs needs extensive review, as its English is markedly poorer than the other parts of the manuscript, and at some points it is insufficiently appropriate. E.g. Line 373: recognition of pattern antigens is false, TLR-s do not recognize antigens but rather molecular patterns, Line 374 The sentence starting with B cells are known to... and ends with development of SLE is without any meaning, Line 376 CpG should be made more accurate by writing CpG-DNA, Line 378 the abbreviations should be explained. Even more importantly, almost all the mechanisms presented in section 5.2 are processes involved during normal B-cell activation. Obviously, after the cells have been activated, normally processes that silence the activation cascade should be initiated in order to terminate the immune reaction. If the aim of this section is to elucidate the mechanisms of this silcencing, regulatory processes, these should be highlighted, not the initial activation steps. Again, differences between these Breg-inducing processes between healthy subjects and lupus patients or lupus animal models should be presented. Line 398 "Interestingly, the elevated serum levels of BAFF after several rounds of rituximab-mediated B cell-depletion..." I think that upregulation of BAFF after the elimination of B-cells is a normal feedback mechanism, and this should not be addressed in this paper.
Author Response
Line 188: "The CD23loCD21hiMZ and CD5+ B-1 B cells respond to T cell-independent antigens and undergo polyclonal expansion during multiple stimuli". This is a very general sentence, and no further detailed description of this process is presented, rather a schematic Figure is presented, but it is not clear, from what research data (citations, own previous experiments) does the information presented in Figure 2 comes from. A more clear description of the proposed mechanism and sources of the data is needed, also specifying if these are animal or human data.
Response: We have added the detailed description of MZ and CD5+ B-1 B cells during infection in the revised main text (please see lines 221-223). We also included the key publications on MZ, CD5+ B-1 B cells and Figure 2 into the revised reference list (please refer to reference 12,71 and 72). All line numbers are shown in track changes with ‘All Markup’ view.
Line 226. Is the PIK3CD-E1020K model an SLE model or of some other autoimmune disease? Line 230-231: The first half of the sentence describes the role of IPK3CDE in B-cell differentiation, and the second half concludes that this indicates an important pathogenic roles of innate-like B cells in SLE. The causal relationship is not clear, and further data supporting this important role should be presented. What functional sequelae of this skewed B-cell differentiation have been published, in general, or related to SLE?
Response: We fully agree with the reviewer’s critical comments and apologize for the confusion caused. The PIK3CD-E1020K model represents the symptoms of activated PI3K-delta syndrome (APDS) in human, which is not directly related to the pathogenesis in SLE. Therefore, we have now deleted the relevant text of PIK3CD-E1020K.
Line 274. The section 4.2 contains no SLE-related information, only general knowledge about plasma cell physiology. Since autoantibodies are crucial to the pathogenesis of SLE, it has long been obvious that plasma cells are important players of SLE pathogenesis. The Chapter 4. therefore should contain novel and lupus-related information relevant to the whole concept of the paper, or this should be omitted.
Response: We have adopted the reviewer’s suggestion and now added the genome-wide association studies of SNPs in PRDM1 region that is associated with SLE development in human and further highlighted the effective plasma cell depletion therapy in the treatment of SLE (please see revised text in lines 345-349).
Line 314."CD9 was identified as a unique functional marker for IL-10-producing regulatory B cells". Does it mean that CD9 is a B-reg specific-marker, or it is expressed in a wider variety of cells, but its function is most important in B-regs?
Response: Recent evidence has identified CD9 as a specific marker for IL-10-producing Breg cells and their progenitors within B subsets (please refer to reference 124: Sun, J.; Wang, J.; Pefanis, E.; Chao, J.; Rothschild, G.; Tachibana, I.; Chen, J.K.; Ivanov, II; Rabadan, R.; Takeda, Y., et al. Transcriptomics Identify CD9 as a Marker of Murine IL-10-Competent Regulatory B Cells. Cell Rep 2015, 13, 1110-1117, doi:10.1016/j.celrep.2015.09.070).
Line 358-: The section 5.2 The regulation of Bregs needs extensive review, as its English is markedly poorer than the other parts of the manuscript, and at some points it is insufficiently appropriate. E.g. Line 373: recognition of pattern antigens is false, TLR-s do not recognize antigens but rather molecular patterns,
Response: We apologize for the inaccurate descriptions. We have now revised the Breg section and changed the ‘pattern antigens’ to ‘pathogen-associated molecular patterns (PAMPs)’ in the revised main text (please see line 442).
Line 374 The sentence starting with B cells are known to... and ends with development of SLE is without any meaning,
Response: We have revised this sentence in the relevant text (please see lines 443-444).
Line 376 CpG should be made more accurate by writing CpG-DNA,
Response: We have changed the ‘CpG’ to ‘CpG-DNA’ in the relevant text (please see line 445).
Line 378 the abbreviations should be explained.
Response: We have added ‘non-obese diabetic’ for the abbreviation ‘NOD’ in the newly revised main text (please see line 448) and added‘Myeloid differentiation primary response 88’for the abbreviation ‘MyD88’ in the revised text (please see line 451).
Even more importantly, almost all the mechanisms presented in section 5.2 are processes involved during normal B-cell activation. Obviously, after the cells have been activated, normally processes that silence the activation cascade should be initiated in order to terminate the immune reaction. If the aim of this section is to elucidate the mechanisms of this silcencing, regulatory processes, these should be highlighted, not the initial activation steps. Again, differences between these Breg-inducing processes between healthy subjects and lupus patients or lupus animal models should be presented.
Response: Section 5.2 highlights the key signals during Breg induction. We agree with the expert reviewer’s comments that a shift of B cell activation, B cell costimulation and cytokine signal pathways during SLE pathogenesis may dampen the regulatory functions of Bregs (please see newly revised text in lines 428-429). In addition, we have also described Breg induction between healthy controls and SLE patients (please see revised text in lines 383-392 and 474-481).
Line 398 "Interestingly, the elevated serum levels of BAFF after several rounds of rituximab-mediated B cell-depletion..." I think that upregulation of BAFF after the elimination of B-cells is a normal feedback mechanism, and this should not be addressed in this paper.
Response: In line with the reviewer’s suggestion, we have now deleted the relevant text.
Reviewer 3 Report
Ma et al has written a reasonably comprehensive review on the potential pathophysiologic relationships between age-associated B cells, autoreactive antibody-forming cells as well as Breg cells and systemic lupus erythematosus (SLE). While such relationships appear sketchy currently and they were described in the manuscript in a patchy fashion, it is strongly believed that these cells are important in SLE, at least in terms of the pathogenesis of the condition.
As in the investigation of T cells, data accrued in mouse models have sparingly been able to be translated to human disease, particularly in the realm of therapeutics. Classical examples we have been seeing in our laboratory are those related to T cells. They are quite plastic in terms of phenotype and functionality. For instance, anti-inflammatory Tregs may be endowed with pro-inflammatory Th1/Th17-like function, depending on the inflammatory milieu.
As a review article, expert input, speculation and provision of the direction of future research are paramount. The authors would like to spend a paragraph or two to express their scientifically supported views as to how future research can be conducted to safely address the impact of the various B cell types described in the manuscript. Particularly, similar to the case of T cells, whether the plasticity of Bregs (if present), would impede the therapeutic potential of SLE.
Minor points
A thorough grammar check will improve the readability of the manuscript. A few spelling mistakes are obvious - for example - "secret" should be spelt as "secrete", "expended" should be "expanded". Figure 4 is not informative. The authors should consider removing it.
Author Response
As a review article, expert input, speculation and provision of the direction of future research are paramount. The authors would like to spend a paragraph or two to express their scientifically supported views as to how future research can be conducted to safely address the impact of the various B cell types described in the manuscript. Particularly, similar to the case of T cells, whether the plasticity of Bregs (if present), would impede the therapeutic potential of SLE.
Response: We appreciated the reviewer’s insightful comments. To address the key questions raised by the reviewer, we have added one paragraph in newly revised text (please see lines 529-544). All line numbers are shown in track changes with ‘All Markup’ view.
A thorough grammar check will improve the readability of the manuscript. A few spelling mistakes are obvious - for example - "secret" should be spelt as "secrete", "expended" should be "expanded".
Response: We apologize for the spelling mistakes, which are corrected in the newly revised main text.
Figure 4 is not informative. The authors should consider removing it.
Response: In line with the reviewer’s suggestion, we have now deleted Figure 4.
Round 2
Reviewer 2 Report
I accept the responses and I consider the revised manuscript as suitable for publication.
Author Response
We appreciated the reviewer’s insightful comments and kind response.
Reviewer 3 Report
The additional contents which the authors added are sufficient but there are still a lot of spelling mistakes and grammatical/language usage issues which remain unresolved.
Otherwise, I am fine with the amended contents which have sufficiently addressed my previous concerns.
Author Response
We apologize for the spelling mistakes and grammatical usage issues, which are corrected in the newly revised main text.